# Image-Based Artificial Intelligence Technology for Diagnosing Middle Ear Diseases: A Systematic Review

**DOI:** 10.3390/jcm12185831

**Published:** 2023-09-07

**Authors:** Dahye Song, Taewan Kim, Yeonjoon Lee, Jaeyoung Kim

**Affiliations:** 1Major in Bio Artificial Intelligence, Department of Applied Artificial Intelligence, Hanyang University, Ansan 15588, Republic of Korea; thdekgp99@hanyang.ac.kr (D.S.); taewankim778@hanyang.ac.kr (T.K.); 2Department of Dermatology and Skin Sciences, University of British Columbia, Vancouver, BC V6T 1Z1, Canada; jaykim830@gmail.com; 3Core Research & Development Center, Korea University Ansan Hospital, Ansan 15355, Republic of Korea

**Keywords:** artificial intelligence, automated diagnosis, deep learning, middle ear diseases

## Abstract

Otolaryngological diagnoses, such as otitis media, are traditionally performed using endoscopy, wherein diagnostic accuracy can be subjective and vary among clinicians. The integration of objective tools, like artificial intelligence (AI), could potentially improve the diagnostic process by minimizing the influence of subjective biases and variability. We systematically reviewed the AI techniques using medical imaging in otolaryngology. Relevant studies related to AI-assisted otitis media diagnosis were extracted from five databases: Google Scholar, PubMed, Medline, Embase, and IEEE Xplore, without date restrictions. Publications that did not relate to AI and otitis media diagnosis or did not utilize medical imaging were excluded. Of the 32identified studies, 26 used tympanic membrane images for classification, achieving an average diagnosis accuracy of 86% (range: 48.7–99.16%). Another three studies employed both segmentation and classification techniques, reporting an average diagnosis accuracy of 90.8% (range: 88.06–93.9%). These findings suggest that AI technologies hold promise for improving otitis media diagnosis, offering benefits for telemedicine and primary care settings due to their high diagnostic accuracy. However, to ensure patient safety and optimal outcomes, further improvements in diagnostic performance are necessary.

## 1. Introduction

Otitis media (OM) is a prevalent ailment in children [1], presenting symptoms such as fever, sleep disturbances, and acute infections [2]. This illness significantly affects not only children who experience considerable pain but also their caregivers [3]. OM prevalence is high worldwide, with rates of 9.2% in Nigeria, 10% in Egypt, 6.7% in China, 9.2% in India, 9.1% in Iran, and 5.1–7.8% in Russia [4]. Additionally, the incidence of OM in native Australian children is 90%, the highest worldwide [5]. Prior works have discussed OM diagnosis and treatment methods [6]. If OM is inaccurately diagnosed, it can lead to severe consequences, including hearing loss, cognitive development disorders, unnecessary surgeries, antibiotic overuse, and disease exacerbation [7]. Notably, 80% of OM patients receive antibiotics, leading to potential antibiotic resistance and unnecessary expenses [8]. Therefore, accurate diagnosis is essential to mitigate these side effects and provide effective treatment.

Diagnostic techniques for both acute and chronic middle ear infections have long posed challenges [7]. Infants, in particular, present difficulties due to their narrow external ducts, which, coupled with the presence of earwax, can hinder accurate diagnosis using an ear endoscope alone [9]. Furthermore, in primary clinics and pediatrics, the accuracy of diagnosis tends to be low due to a lack of systematic training and unfamiliarity with pneumatic ear endoscopy [10,11]. To address these challenges, various approaches have been explored in the field. These include specialized training programs for medical students, the development of new otoscopic approaches and techniques, the implementation of absorbance and acoustic admittance measurements, and the integration of impedance-measuring hearing aids. Additionally, clinical trials have been conducted to compare the effects of these various approaches [7]. However, despite these approaches and efforts, the diagnostic success rates among pediatricians and otolaryngologists in primary care settings do not exceed 70% [7].

Medical image processing is of considerable importance in the analysis and exploration of medical data [12]. However, the complexity inherent in medical images presents challenges to their accurate representation and evaluation using conventional approaches. The use of AI has demonstrated a high level of effectiveness in the analysis of these complex medical images [13], which has led to its frequent use in medical research [14]. Diagnostic accuracy in otolaryngology can vary based on a physician’s training and area of specialization, given the reliance on endoscopic imaging and visual mechanisms [15]. Therefore, the integration of deep learning algorithms in oto-endoscopic imaging is of significant importance. Due to advances in computer science, the utilization of AI in the medical field has seen substantial growth, particularly in studies involving endoscopic images [16,17]. Despite this progress, the application of an automatic diagnostic system for OM in actual clinical settings remains unimplemented due to uncertainties associated with deep learning, posing a major obstacle as identified in reference [18].

In this study, we evaluated the diagnostic accuracy according to the AI technology used in automatic OM diagnostic studies based on medical images and the type of OM diagnosed. Based on the findings of the study, we discuss improvement measures in this paper and suggest directions for future research.

## 2. Materials and Methods

### 2.1. Search Strategy

This review explores the use of AI in the study of middle ear disease. We specifically examined: (1) automated diagnostic systems utilizing artificial intelligence (AI) based on medical imaging and (2) middle ear disease. In addition to the reviewed studies, the materials from existing survey studies [17,19] were also compiled. Literature searches were conducted on Google Scholar, PubMed, Medline, Embase, and IEEE Xplore databases, using a combination of otitis-media-related and AI-related keywords. Otitis-media-related keywords included ‘otitis media’, ‘ear abnormalities’, ‘ear pathology’, ‘tympanic membrane disease’, ‘otorhinolaryngology’, ‘middle ear’, and ‘eardrum’. AI-related keywords included ‘artificial intelligence’, ‘machine learning’, ‘deep learning’, ‘automation’, ‘computer diagnostics’, ‘diagnose’, ‘convolutional neural networks’, ‘neural networks’, ‘classification’, ‘segmentation’, ‘supervised learning’, and ‘unsupervised learning’. The AND and OR operators were employed to explore various keyword combinations.

### 2.2. Article Appraisal Method

The literature from Google Scholar was collected using the ‘Publish or Perish version 8’ software. ‘EndNote version X9’ was utilized to eliminate duplicates from the collected works. Subsequently, a detailed review of titles and abstracts was undertaken to identify and analyze the gathered literature.

### 2.3. Inclusion and Exclusion Criteria

Resources were sought without temporal restrictions, aiming to identify AI technology applied in the diagnosis of OM. Studies focusing on the diagnosis of OM without the inclusion of AI, those not written in English, or those lacking the full text were excluded. The selection and review process of the literature followed the PRISMA criteria (Figure 1) [20]. Discussions and agreements during the screening process were conducted in consultation with all authors. We primarily reviewed publications that focused on AI technology based on medical imaging. Among the collected documents, we filtered out survey documents, not research ones, and added documents that were included in these survey publications but were not included in the literature review process.

## 3. Result

Medical imaging studies on middle ear diseases can be categorized into those employing classification, segmentation, and a combination of both techniques, as illustrated in Figure 2. Our review included a total of 32 papers, comprising 26 utilizing classification, 3 applying segmentation, and 3 using both approaches. Classification-based studies, which formed the majority, typically relied on images of the tympanic membrane for disease diagnosis. In the process of diagnosis, a simple classification may not provide the cause of the disease, that is, it may not identify the ‘where’ of the problem [21]. Moreover, as otolaryngology diagnoses are typically carried out through endoscopy, variation is inevitable, contingent on the individual conducting the diagnosis. In order to avoid subjective bias in otoscopy examinations and to enhance diagnostic accuracy, a model capable of interpreting the structure of the tympanic membrane through endoscopic images is required [21]. Consequently, research has emerged focusing on the segmentation of tympanic membrane images in detail. There have also been studies that combined segmentation—a detailed division of the tympanic membrane structure—with classification. It has been observed that the average diagnostic performance of studies that incorporated segmentation with classification was higher than that of studies that only performed classification [22]. In this section, we present the surveyed papers, categorized as classification, classification and segmentation, and segmentation.

### 3.1. Classification

Image classification serves as a fundamental approach within the realms of computer vision and pattern recognition [23]. Oto-endoscopic, CT, and smartphone-based low-cost sword mirrors are among the forms of data used for classification in otorhinolaryngology, as presented in Table 1. Furthermore, the number of classification labels ranges from 2 to 14. Classification methods include machine learning methods that extract features of images and apply them to classifiers and methods using convolutional neural networks (CNN) models of deep learning, which are typically used for image classification.

Firstly, an example of a method employing machine learning is observed in the study by Hermanus et al., where they expanded the automatic diagnostic system for middle ear diseases into an internet-connected Android smartphone-based system [24]. In a context where most developing countries suffer from limited access to medical care, leading to a rise in the prevalence of ear diseases, as noted by Ibekwe et al. [46], the significance of such a development is amplified. This study emphasized the challenges faced by countries with deficient medical technology, particularly a lack of sufficient experience in diagnosing ear diseases. To address this, an effective, low-cost otoscope-based automatic ear disease diagnostic smartphone application was implemented, and its efficacy was verified. Utilizing 389 video-otoscope images, they designed a system that diagnoses five conditions—Normal, obstructing wax or foreign bodies in the external ear canal (W/O), acute otitis media (AOM), otitis media with effusion (OME), and chronic suppurative otitis media (CSOM)—automatically through a low-cost otoscope. Image pre-processing was initially performed via cropping and blur detection. The application of a decision tree to feature vectors, extracted through feature extraction processes including color detection, edge detection, blob detection, and shape detection, facilitated disease classification with an accuracy of 81.58%. Furthermore, the authors separately developed a classification model by training a neural network comprising seven input layers, ten hidden layers, and five output layers. Following the adaptation of this model into a smartphone application with a portable otoscope, it demonstrated a diagnostic accuracy of 86.84%, a rate that holds up impressively against the diagnostic accuracy of an otolaryngology expert.

Within the 26 articles focused on image classification, 17 employed deep learning methodologies, specifically CNN. Notably, the study with the most expansive classification of diagnoses utilized a DensNet-BC169 and DensNet-BC1615-based ensemble classifier. Applied to a substantial dataset of 20,542 endoscopic images, this classifier distinguished eight types of middle ear diseases, achieving an impressive accuracy of 95.59% [25].

In another study, CNN models were implemented into a web-based program. Khan et al. developed a model to identify chronic otitis media (COM) with perforation and OME in oto-endoscopic images (OEIs), designed to assist otolaryngologists in primary care settings [26]. Hughes and colleagues highlighted the significant disparity between the number of otolaryngologists and the general population in the United States [47]. Furthermore, a study by Pichichero et al. pointed out the low diagnostic accuracy among pediatricians (50%) and otolaryngologists (73%) and suggested the utilization of additional diagnostic tools to enhance the accuracy of otitis media diagnosis [11]. Motivated by these findings, Khan et al. tested several renowned CNN architectures (ResNet50, DenseNet161, VGGNet16 (BN), Inception-ResNet v2, and SE ResNet152), following the collection and augmentation of 2484 otoscopic images. The resulting diagnostic model for middle ear disease, which employed DenseNet161 due to its superior performance, achieved an accuracy of 95%. They also utilized Grad-CAM to validate whether the model’s disease classifications were based on medically reasonable areas in the imaging study. It was confirmed that the model’s classification mechanism mirrored the visual diagnostic approach of a medical expert. By creating a web-based otolaryngology evaluation system and juxtaposing its performance with that of actual otolaryngology experts (comprising seven specialists, six residents, and four interns), the model demonstrated superior diagnostic accuracy in the evaluation system.

Since the discovery of X-ray radiation, substantial advances have been made in the field of medical imaging, with technologies such as Computed Tomography (CT) and Magnetic Resonance Tomography (MRT) facilitating diagnostic and treatment planning processes beyond the capabilities of traditional imaging modalities [12]. For example, Eroğlu et al. implemented an AI model that automatically diagnoses the presence or absence of cholesteatoma in COM [27]. From an anatomical perspective, OM exhibits varying degrees of damage. Particularly in cholesteatoma, larger and more extensive bone damage is observed. The researchers emphasized the need for the rapid diagnosis and treatment of COM with cholesteatoma, as bone damage is a significant factor causing not only temporal bone and intracranial complications but also conductive or sensorineural hearing loss. This research utilized 3,093 CT images, recorded in JPEG format, including the middle ear and temporal bone. The deep learning models used were AlexNet, GoogLeNet, and DenseNet201. Subsequently, they combined the three feature maps derived from each model using a Support Vector Machine (SVM) to categorize into Normal, Cholesteatoma with COM, and Cholesteatoma without COM. The diagnostic accuracy demonstrated an excellent result of 95.4%.

Research has also incorporated ensemble learning, which combines the results of multiple CNN models, rather than relying on a single CNN model. Cha et al. attempted various diagnoses of middle ear diseases using artificial intelligence [28]. The early and accurate diagnosis of middle ear diseases is essential, but this proves difficult in developing countries. Pichichero et al. showed that the diagnostic accuracy of pediatricians for middle ear diseases is only about 50%, while otolaryngologists also demonstrated an unsatisfactory diagnostic accuracy of 73% [11]. Furthermore, the best-known study among auto-diagnosis papers of middle ear diseases using oto-endoscopic images so far showed an accuracy of 86.84%, but it only partially diagnosed OM. Thus, this research diagnosed not only OM but also included a broader range of middle ear diseases. They utilized 10,544 otoscopic images, classifying them into six categories. The normal category included completely normal tympanic membranes, those that appear normal or show healed perforations, and tympanosclerosis. The five abnormalities categorized were tumors (middle ear tumors, EAC tumors, cerumen impaction), OME, eardrum erosions and otitis externa, perforation of the eardrum, and attic retraction/atelectasis. The classified data were trained on AlexNet, GoogLeNet, ResNet (ResNet18, ResNet50, ResNet101), Inception-V3, Inception-ResNet-V2, SqueezeNet, and MobileNet-V2. Selecting the two models (Inception-V3, ResNet101) that demonstrated the best performance and creating an ensemble classifier resulted in a diagnostic accuracy of 93.67%.

Upon comparing the overall trends and results of these studies, it was observed that most research utilizing CNN models demonstrated higher classification accuracy. Additionally, studies that employed a greater number of images for training generally yielded better results.

### 3.2. Segmentation

Image segmentation is a procedure for the extraction of ROIs from an image using an automatic or semi-automatic process [48]. A medical image is crucial because clinicians utilize it to investigate the anatomical structure of patients. In medical applications, numerous image segmentation methods have been utilized to segment tissues and organs. In otolaryngology, endoscopic image data are typically used for segmentation, and segmentation studies include the diseased tympanic membrane to diagnose middle ear illnesses or the interior structure of the tympanic membrane, vertebrae, and middle ear in detail. In the studies included in our investigation, as shown in Table 2, the methods of segmenting medical images used mask R-CNN models and UNet-based models.

Pham and colleagues presented a method for the complete automatic segmentation of the tympanic membrane, which includes the disease [49]. Thus far, various techniques have been developed for tympanic membrane segmentation. Hsu et al. [51], Ibekwe et al. [52], and Ribeiro et al. [53] proposed a semi-automatic method for segmenting the tympanic membrane, in which the ROI is marked manually with a mouse. Xie et al. [54], Shie et al. [22], and Tran et al. [2] implemented a contours models. These methods, as noted by the authors, may produce unsatisfactory results due to variability among individuals and when the boundaries of the input images are weak or damaged by noise. Therefore, Pham et al. employed a CNN-based segmentation model for the consistent automatic segmentation of the tympanic membrane [49]. In their study, they developed an EAR-Unet model, applying EfficientNet-B4 to the Unet encoder, ResNet Block to the decoder, and the Attention gate to the skip connections. The EAR-Unet was applied to 1012 otoscopic images to segment the tympanic membrane in states of Normal, AOM, COM, and OME. When compared with existing segmentation models such as FCN, SegNet, Unet, Attention Unet, and Residual Unet, the EAR-Unet demonstrated the best performance with an accuracy of 95.8%.

In their research targeting the detailed segmentation of the inner structures of the tympanic membrane, Seok et al. developed a deep learning model to identify and segment key structures [21]. They pointed out that existing deep learning and machine learning techniques, despite having been used for the automatic diagnosis of middle ear diseases, may not be sufficiently applicable in actual clinical situations. The authors expressed concern about the potential decrease in credibility of methods that only classify a tympanic membrane image as diseased. This is mainly due to the absence of explanations addressing the ‘why’ and ‘where’ aspects of the disease manifestation. Consequently, they highlighted the necessity of a deep learning model that interprets tympanic membrane images based on specific structures or findings and provides useful results. For this purpose, they augmented 920 endoscopic images, labeled using the LabelMe application, and employed a Mask R-CNN Segmentation model with ResNet-50 as the backbone. This approach allowed them to determine and segment the tympanic membrane, malleus with side of tympanic membrane, and presence or absence of the perforations. Using the Intersection over Union (IoU) evaluation metric, they verified the potential of the deep learning model for detecting and segmenting major structures in oto-endoscopic images, achieving 100% for the tympanic membrane, 88.6% for the tympanic membrane based on the malleus, and 91.4% for perforations.

### 3.3. Classification and Segmentation

Research efforts that incorporate segmentation images into classification aim to enhance diagnostic performance. This is achieved by extracting the ROI and focusing on distinguishing features during disease classification. As shown in Table 3, this approach has been primarily used on tympanic membrane otoscopic images in the field of otolaryngology. The common methodology involved detecting the tympanic membrane area within the entire endoscopic image, eliminating all other portions, and subsequently inputting the resulting image into the classification model. These classification models varied, with some studies utilizing CNN models and others employing feature-based classifications that extracted specific characteristics from the images.

In their research aimed at diagnosing OM by segmenting the tympanic membrane and using machine learning, Shie and colleagues proposed a novel hybrid OM CAD system for automatically diagnosing various forms of OM [22]. Teele et al. stated that AOM is one of the most common reasons for visits to pediatricians, and that it incurs significant costs and causes substantial social burden, leading to large indirect losses each year [57]. The diagnosis of diseases such as AOM in children, OME, and COM is complex due to unclear or varied symptoms, necessitating an OM CAD system that can assist regions with limited medical resources. The research conducted by the authors involved several steps. Firstly, they proposed the use of the double active contour method for segmenting the tympanic membrane from 865 otoscopic images. Following this segmentation, they proceeded to utilize an SVM classifier with Adaboost applied, aiming to distinguish between the Normal, AOM, OOME, and COM conditions. Initially, the bright area of the external ear canal (where light is reflected) was eliminated using the active contour. Then, the tympanic membrane was segmented into a shape resembling a circle or ellipse. For diagnosing middle ear diseases, the visual characteristics of each disease were extracted from the previously segmented tympanic membrane images by applying GCM, HOG, LBP, and Gabor filters, and then converted into feature vectors. Afterwards, the authors applied Adaboost to an SVM classifier to distinguish Normal, AOM, OME, and COM with a diagnostic accuracy of 88.06%. The study showed higher classification accuracy when using tympanic membrane images segmented with the double active contour method compared to the original images. However, due to the loss of important features of the tympanic membrane during the segmentation process, Shie and colleagues evaluated that the improvement in accuracy was lower than expected.

As an example of using a CNN model for classifying segmented images, there is a study by Başaran et al. in which they segmented the tympanic membrane to diagnose middle ear diseases [55]. Currently, visual inspections of the tympanic membrane and ear canal are conducted in hospitals. Pichichero et al. argued that this approach is not objective due to the variability of observations during diagnoses and the inclusion of human errors [11]. Goggin et al. also pointed out that the use of computer-aided diagnostics or expert systems is limited in the field of otolaryngology [58]. To overcome these issues and to facilitate objective inspections, this study used a CNN model for the automatic detection and classification of the tympanic membrane. They utilized 282 augmented otoscopy images, which included Normal, AOM, Earwax, Myringosclerosis, Tympanostomy tubes, CSOM, and Otitis externa. They used a fine-tuned Faster R-CNN for tympanic membrane detection and trained models such as AlexNet, Vgg-16, Vgg-19, GoogLeNet, ResNet50, and ResNet101 for disease classification. Among these models, Vgg-16 demonstrated the highest performance with a diagnostic accuracy of 90.48%.

## 4. Discussion

In this review, we investigated the application of AI technology for the detection and diagnosis of middle ear diseases using endoscopic imaging. AI technology was divided into three categories: classification studies, segmentation studies, and classification and segmentation studies. Classification studies had an average diagnosis accuracy of 86%, with maximum accuracy of 99.16% and minimum accuracy of 48.7%. Classification with segmentation studies had an average diagnosis accuracy of 90.8%, with maximum accuracy of 93.9% and minimum accuracy of 88.06%. The average diagnosis accuracy of all investigated studies was 86.5%, which was higher than the diagnostic accuracy of pediatricians and otolaryngologists in primary care (70%). This indicates that AI studies utilizing medical imaging could aid primary care in otolaryngology [7].

The diagnostic accuracy of studies employing CNN models to diagnose middle ear diseases was found to be approximately 6% higher than that of studies that did not utilize CNN models (e.g., studies using classifiers for feature vectors and studies of applied CBIR systems). Among the studies that utilized CNN models, a study compared the classification results of models to those of actual specialists, and the models correctly diagnosed middle ear disease with 95% accuracy [26]. In addition, Byun et al. diagnosed ear disease with a high accuracy of 97.18% and confirmed that specialists’ diagnostic accuracy improved by up to 18% (1.4–18.4%) when using the proposed model in actual clinical situations [30]. Furthermore, telemedicine systems incorporating AI technology with accuracy comparable to that of a specialist could be beneficial for patients in areas with a shortage of specialists or for those who find it difficult to visit a hospital [59]. Myburg et al. established an android telemedicine system capable of classifying images based on feature vectors for disease diagnosis [24]. They achieved a diagnostic accuracy of 81.5% using a decision tree model which improved to 86.84% accuracy when utilizing self-implemented deep neural network (DNN) models.

Currently, it is difficult to apply medical AI to actual clinical situations owing to a number of drawbacks, including data scarcity, inapplicability outside of the training domain, and misdiagnosis due to data imbalance [60]. The literature analyzed in this study indicates a higher diagnostic accuracy than that of primary care specialists, but it is predominantly based on supervised learning, making it less applicable to real-world clinical settings where cases are more diverse. In addition, it was determined that the accuracy of the studies applied to a remote diagnostic system was lower than the accuracy of the research as a whole. In addition, because the performance of the model depends on the quantity and quality of the data, the diagnostic accuracy between studies varies considerably (48.7–99.16%). In future research, it is essential to pay attention to the quality and quantity of images for consistent results. For instance, earwax can obstruct an otolaryngologist’s view of the eardrum in real clinical situations, and this also applies to images used for artificial intelligence. Notably, in pediatric diagnoses, it is challenging to identify otitis media due to the presence of earwax [61,62]; therefore, to ensure high performance, it is necessary to remove earwax before capturing endoscopic images.

## 5. Conclusions

This paper aimed to examine the possibility of developing AI research in otolaryngology using medical imaging. Otoendoscopy image-based studies were the most prevalent and showed high accuracy. Medical images, such as MRI and CT, are not frequently utilized for the automatic diagnosis of middle ear diseases. The various studies aimed at OM classification, segmentation, and classification with segmentation, with classification research comprising the majority of these studies. The results of these studies varied depending on the amount of image data collected, the number of classes, and the type of model. Usually, the results of pre-trained models are superior, and the smaller the number of classes, the greater the accuracy. High diagnostic accuracy was also demonstrated when the number of classes was large, but the collected image data were large. Some studies have suggested methods for enhancing precision using ensemble learning with models or segmenting data.

In conclusion, the quality of AI research in otolaryngology that utilizes medical images will continue to improve. Future work is required for the above technologies to be applicable in real clinical settings.

## Figures and Tables

**Figure 1 jcm-12-05831-f001:**
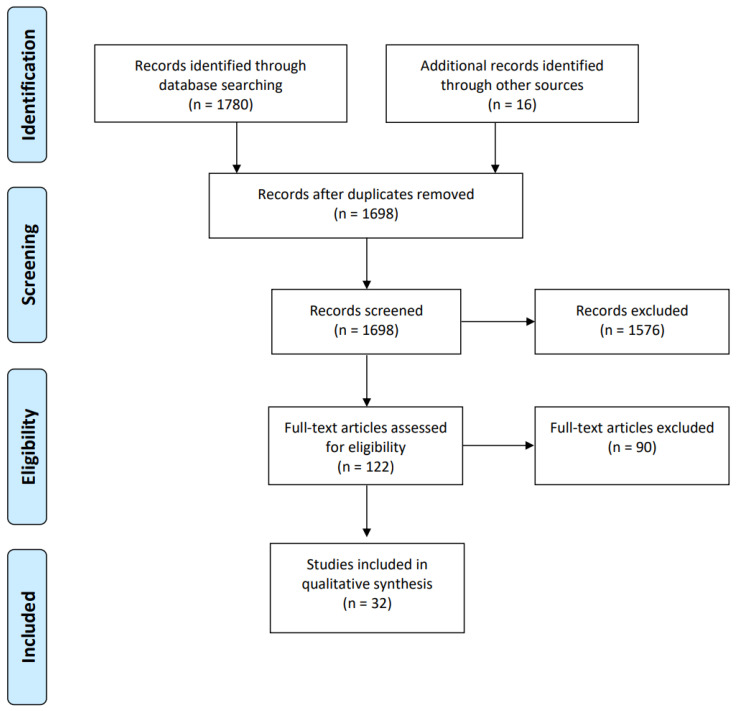
Overview of review-based PRISMA guide (Appendix A).

**Figure 2 jcm-12-05831-f002:**
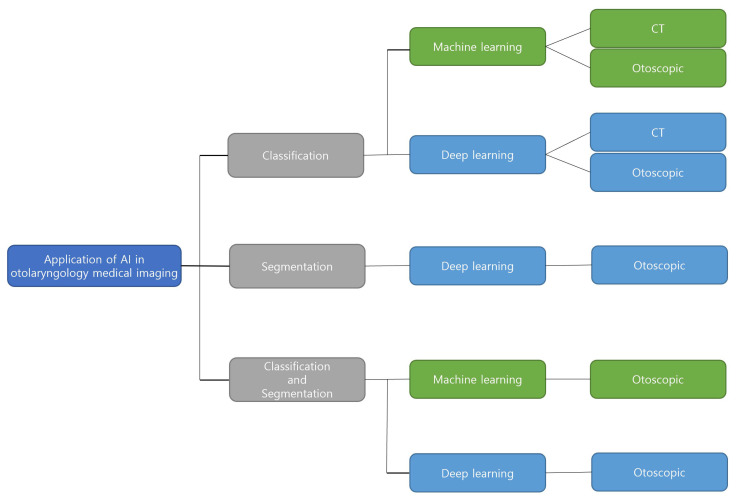
Overview of artificial intelligence studies applied to middle ear image.

**Table 1 jcm-12-05831-t001:** Overview of classification studies.

No.	Author	Number of Classes	Model	Image Type	Number of Images	Outcomes
1	Tran, T. et al. (2018) [2]	2	Multitask joint sparse representation-based classification (MTJSRC)	Otoscopic	214	Accuracy: 91.41%
2	Crowson, M.G. (2021) [7]	2	ResNet-34	Endoscopic	338	Accuracy: 83.8%
3	Wu, Z. et al. (2021) [8]	3	Xception	Otoendoscopy	12,203	Accuracy 97.45%
4	Monroy, G.L. (2019) [15]	3	Twenty-two classifiers in MATLAB, random forest classifier	Optical coherence tomography (OCT)	25,497	Accuracy: 99.16%
5	Myburgh, H.C. et al. (2018) [24]	5	Neural network, Decision tree	Commercial video-otoscopes	389	Neural network accuracy: 86.84%,Decision tree accuracy: 81.58%
6	Zeng, X. et al. (2021) [25]	8	DenseNet169, DenseNet1615	Endoscopic	20,542	Accuracy: 95.59%
7	Khan, M.A. et al. (2020) [26]	3	DenseNet161	Otoendoscopy	2,484	Accuracy: 94.9%
8	Eroğlu, O. et al. (2022) [27]	3	(Alexnet, Googlenet, Densenet201) + SVM	CT	3093	Accuracy 95.4%
9	Cha, D. et al. (2019) [28]	6	InceptionV3, ResNet101	Otoscopic	10,544	Accuracy: 93.73%
10	Habib, A.R. et al. (2020) [29]	4	InceptionV3	Otoscopic	233	Accuracy: 76.0%
11	Byun, H. et al. (2021) [30]	4	ResNet18 + Shuffle	Endoscopic	2272	Accuracy: 97.18%
12	Mironică, I., Constantin, V., Dan, C.G. (2011) [31]	2	Neural Networks	Otoscopic	186	Accuracy: 73.11%
13	Wang, X., Tulio, A.V., Jinbo, B. (2015) [32]	2	cascaded classifier, SVM	Otoscopic	215	Accuracy: 90%
14	Myburgh, H.C. et al. (2016) [33]	5	Decision tree	Commercial video-otoscopes, Low cost custom-made video-otoscope	489	Commercial video-otoscopes accuracy: 80.6%, Low cost custom-made video-otoscope accuracy: 78.7%
15	Lee, J.Y., Choi, S., Chung, J.W. (2019) [34]	2, 2	Neural Networks	Endoscopic	1338	Tympanic membrane direction Accuracy: 97.9%, Perforation Accuracy: 91.0%
16	Livingstone, D. et al. (2019) [35]	3	Neural Networks	Otoscopic	734	Accuracy: 84.4%
17	Başaran, E. et al. (2019) [36]	2	Gray-level co-occurrence matrix (GLCM) and artificial neural network (ANN)	Otoscopic	223	Accuracy: 76.14%
18	Livingstone, D., Justin, C. (2020) [37]	14	Multilabel classifier architecture	Otoscopic	1366	Accuracy: 88.7%
19	Camalan, S. et al. (2020) [38]	3	Content-based image retrieval (CBIR) system	Otoscopic	454	Accuracy: 80.58%
20	Won, J. et al. (2021) [39]	2	Random forest	A-scan OCT	25,479	Accuracy: 91.5%
21	Tsutsumi, K. et al. (2021) [40]	5	MobileNet-V2	Otoscopic	400	Accuracy: 77.0%
22	Sundgaard, J.V. et al. (2021) [41]	3	inceptionV3	Otoscopic	1,336	Accuracy: 86%
23	Singh, A. and Malay, K.D. (2021) [42]	4	Neural Networks	Otoscopic	880	Accuracy: 96%
24	Miwa, T. et al. (2022) [43]	3	Single Shot MultiBox Detector (SSD)	CLARA + CHROMA, SPECTRA A, SPECTRA B	826	Accuracy: 48.7%
25	Binol, H. et al. (2022) [44]	4	OtoXNet	Otoscopy	765	Accuracy: 84.8%
26	Habib, A. et al. (2022) [45]	5	ResNet backbone	Endoscopic	6,527	Accuracy: 74.5%

**Table 2 jcm-12-05831-t002:** Overview of segmentation studies.

No.	Author	Model	Image Type	Number of Images	Outcomes
1	Seok. J. et al. (2019) [21]	Mask R-CNN (ResNet-50 backbone)	Endoscopic	920	Accuracy: 92.9%
2	Pham. V. et al. (2021) [49]	EAR-UNet	Otoscopic	1012	Accuracy: 95.8%
3	Binol. H. et al. (2020) [50]	UNet	Otoscopic	900	Kendall’s Coefficient: 83.9%

**Table 3 jcm-12-05831-t003:** Overview of classification and segmentation studies.

No.	Author	Number of Classes	Model	Image Type	Number of Images	Outcomes
1	Shie, C. et al. (2014) [22]	4	Segmentation: Active Contour Models, Classification: Adaboost (adaptive boosting)	Otoscopic	865	Accuracy: 88.06%
2	Başaran, E., Zafer, C., Çelik, Y. (2020) [55]	7	Segmentation: Faster R-CNN, Classification: Vgg16	Otoscopic	282	Accuracy: 90.45%
3	Viscaino, M. et al. (2020) [56]	4	Segmentation: Hough Transform, Classification: SVM	Otoscopic	720	Accuracy: 93.9%

## Data Availability

No new data were created or analyzed in this study.

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
