# Peer review of "Image-Based Artificial Intelligence Technology for Diagnosing Middle Ear Diseases: A Systematic Review"

_jcm, 2023, doi:10.3390/jcm12185831_

Round 1
Reviewer 1 Report
In this review, the application of AI technology for the detection and diagnosis of middle ear diseases using endoscopic imaging is evaluated in three stages within the scope of classification, segmentation studies and classification and segmentation studies.
In the diagnosis of middle ear diseases, more contributions can be made to artificial intelligence modules instead of medical images such as MR and CT with such compilations. This review is sufficient for this subject in terms of methods and results.
English language is enough.
Reviewer 2 Report
The subject is of utmost importance. Therefore I congratulate the authors for their paper.
I have some minor remarks: Using otoendoscopy at least the otologist often is confronted with ear wax which means an obstacle to obtain a proper view of the ear drum. Therefore a meticulous cleaning of the external ear canal is mandatory before endoscopy can be performed. Perhaps the authors can discuss this difficulty in daily practice.
The authors mention the LVAS; this is no disease of the middle ear.
Reviewer 3 Report
Overall, I think this was an excellent paper that was well written, interesting, and would be useful for the otolaryngology community. My one major criticism is simply that the discussion could be longer, and the conclusion could be shorter. I think most of the conclusion other than the last two sentences would fit better in the discussion. Finally, I think the discussion could be more thorough and perhaps give a glance into the authors' opinions on AI and its future directions, etc., moving forward in otolaryngology.
